# Development of Bisphenol-A-Glycidyl-Methacrylate- and Trimethylolpropane-Triacrylate-Based Stereolithography 3D Printing Materials

**DOI:** 10.3390/polym14235198

**Published:** 2022-11-29

**Authors:** Yura Choi, Jisun Yoon, Jinyoung Kim, Choongjae Lee, Jaesang Oh, Namchul Cho

**Affiliations:** 1Department of Energy Systems Engineering, Soonchunhyang University, Asan 31538, Republic of Korea; 2Department of Neurosurgery, College of Medicine, Soonchunhyang University, Asan 31538, Republic of Korea

**Keywords:** 3D printing, photocurable polymer, acrylate polymer, mechanical properties, stereo-lithography

## Abstract

The main advantages of the three-dimensional (3D) printing process are flexible design, rapid prototyping, multi-component structures, and minimal waste. For stereolithography (SLA) 3D printing, common photocurable polymers, such as bisphenol-A glycidyl methacrylate (Bis-EMA), trimethylolpropane triacrylate (TMPTMA), as well as urethane oligomers, have been widely used. For a successful 3D printing process, these photocurable polymers must satisfy several requirements, including transparency, a low viscosity, good mechanical strength, and low shrinkage post-ultraviolet curing process. Herein, we investigated SLA-type photocurable resins prepared using Bis-EMA, TMPTMA, and urethane oligomers. The flexural strength, hardness, conversion rate, output resolution, water absorption, and solubility of the printed materials were investigated. The degree of conversion of the printed specimens measured by infrared spectroscopy ranged from 30 to 60%. We also observed that 64–80 MPa of the flexural strength, 40–60 HV of the surface hardness, 15.6–29.1 MPa of the compression strength, and 3.3–14.5 MPa of the tensile strength. The output resolution was tested using three different structures comprising a series of columns (5–50 mm), circles (0.6–6 mm), and lines (0.2–5 mm). In addition, we used five different pigments to create colored resins and successfully printed complex models of the Eiffel Tower. The research on resins, according to the characteristics of these materials, will help in the design of new materials. These results suggests that acrylate-based resins have the potential for 3D printing.

## 1. Introduction

Three-dimensional (3D) printing is a powerful method that is used to create highly complex and multicomponent structures with a well-defined design and composition. As an additive manufacturing (AM) technology, 3D printing is uses 3D model data to create diverse structures and complex shapes. The AM process has attracted considerable interest in academic research and industry. Charles Hull invented the stereolithography (SLA) version of 3D printing technology in 1986 [1]. SLA 3D printing later led to the development of fused deposition modeling, powder bed fusion, contour crafting, and inkjet printing. Because of its rapid and cost-effective prototyping capabilities, 3D printing was widely used by architects, designers, and artists in the early stages of its development to produce aesthetic and functional prototypes [2]. New applications have emerged because of the development of new materials and AM technologies. Technology advances have reduced the cost of 3D printers, extending their applications to schools, laboratories, homes, and small businesses [3]. Furthermore, 3D printing technology is gaining widespread attention in the medical community owing to its ability to produce various medical implants from tissue replicas obtained using computed tomography images [4,5].

Three-dimensional printing uses a variety of materials, including polymers, metals, and ceramics. The primary polymers used in 3D printing are composites of acrylonitrile butadiene styrene and polylactic acid. Advanced metals and alloys are frequently used in aerospace applications [6,7]. However, owing to the poor mechanical properties and anisotropic behavior of 3D-printed products, large-scale printing is not feasible. Therefore, 3D printing optimization is essential for controlling defect sensitivity and anisotropic behavior. Furthermore, changes in the printing environment may affect the quality of the printed products [8].

Among the existing 3D printing technologies, SLA was one of the first AM methods, developed in 1986. In SLA, ultraviolet (UV) light (or an electron beam) is used to initiate a chain reaction in a layer of resin or monomer solution. The quality of the products manufactured using SLA printing technology may vary depending on the manufacturing environment and the chemical and mechanical properties of the material. Polymers are the most commonly used materials in SLA printers. Polymers generally have the mechanical properties of ultimate elongation and strength, and thermal properties of heat resistance and flame retardancy [9,10]. Therefore, it is necessary to control the conditions for optimizing 3D printing technology. Material control is more effective than environmental control. High-quality products can be obtained by selecting and using materials with a high stability and physical resistance [11,12].

The polyurethane acrylate series is an acrylic polymer with an excellent flexibility, low-temperature properties, curing speed control, cohesiveness, and chemical resistance [13,14]. Polyurethane acrylate is characterized by double bonds with a high reactivity at both ends. Depending on the composition and properties of the acrylic series used for synthesis, various structures and properties can be obtained [12,13,15,16,17,18]. The most commonly used monomers for 3D printing resins are bisphenol-A-glycidyl dimethacrylate (Bis-GMA), urethane dimethacrylate (UDMA), bisphenol-A-ethoxylated-glycidyl dimethacrylate (Bis-EMA), and trimethylolpropane triacrylate (TMPTMA) [13,19,20]. However, in the case of dimethacrylate monomers, the high molecular weight and viscosity may reduce the 3D printing resolution [21]. Bisphenol A methacrylate (Bis-EMA) and TMPTMA were used as low-viscosity hydroxylated monomers. The hydrophobic monomers Bis-GMA and Bis-EMA have similar molecular structures. Strong hydrogen bonding to the hydroxyl functional groups of the Bis-GMA structure increases the viscosity of the resin. Therefore, it should be combined with low-concentration monomers, such as TMPTMA [22,23,24]. Furthermore, 3D printer resins contain materials with a variety of chemical properties and components, such as triethyleneglycol dimethacrylate (TEGDMA), tetraethyl orthosilicate (TEOS), and 3-(trimethoxysilyl) propyl methacrylate [25].

Depending on the application, a variety of pigments are used in 3D printing resins. The pigments used in 3D printer resins are classified as organic or inorganic pigments. In addition to color, the pigments are also classified based on the particle size and degree of dispersion. Pigments play the role of additives in 3D printer resins. Therefore, the properties of the additive may affect the control of the resin’s properties [26]. Therefore, an optimal ratio of the polymers must be used to develop a material suitable for the characteristics and performance of a particular 3D printer resin [27,28]. Many parameters, such as the mechanical properties (flexural strength, hardness, etc.) and physical properties (viscosity, color, etc.), must be considered in the selection and formulation of 3D printing resin materials and additives [29,30,31].

In this study, we investigated how the chemical and mechanical properties of 3D printing resins were affected by the mixing ratio of several different acrylate monomers. The viscosity, degree of conversion (DC), and mechanical properties were analyzed and correlated with the proportions of Bis-EMA, TMPTMA, and urethane oligomers. The optimized 3D printing resin exhibited an excellent printing performance, with a high resolution. The resolution of the Shindoh 3D printer was approximately 0.1 mm. Furthermore, a good resolution was achieved with the studied resin for the following test patterns: 5 mm for the column patterns, 600 μm for the circle patterns, and 200 μm for the line patterns. These resolution results indicate that the developed resins are optimal for use with a Sindoh A+ 3D printer. The resin applied in this study showed satisfactory results in terms of its both physical and mechanical properties, as well as the output resolution. These results show various possibilities and applications of 3D printing resins. A resolution of 200 μm or less is required in the field of dental and medical modelling resins that require sophisticated and high mechanical properties. Basic research on the physical properties of resins is very important for obtaining an μm-scale resolution in the use of 3D printers. The resin used in this study showed an output resolution of less than 200 μm. Basic research on the compositions of these resins provides an opportunity to improve the performance of 3D printer resins and compensate for their weaknesses. In addition, it can provide ideas for the use of additives, such as pigments and ceramics, and contribute to the development of 3D printer materials. Thus, we compared the resolutions of complex structures using various pigment additives in the range of 100–200 μm. We observed that all the resins showed a good output resolution. Therefore, the performance of these resins indicates their potential applications in industrial fields requiring a high mechanical strength, dental and medical fields sensitive to color stability and moisture affinity, and artistic fields requiring various pigments [32].

## 2. Materials and Methods

### 2.1. Materials

The SLA 3D printer A+ was obtained from Shindoh Co., Ltd. (Seoul, Republic of Korea), and the UV-light Cure Unit was equipped with a post-curing device (Korea).

Figure 1 depicts the chemical structures of the acrylate based monomers and the photo-initiator. Bis-EMA and TMPTMA were purchased from MIWON Chemical Co., Ltd. (Gyeonggi-do, Republic of Korea). Diphenyl (2,4,6-trimethyl benzoyl) phosphine oxide (TPO), used as a photo-initiator, was purchased from MIWON Chemical Co., Ltd. (Gyeonggi-do, Republic of Korea). The urethane acrylate copolymer used as the oligomer was obtained from Asan Materials Co., Ltd. (Asan, Republic of Korea). = 2,2′-[(3,3′-Dichloro [1,1′-biphenyl]-4,4′-diyl)bis(azo)]bis[N-(4-chloro-2,5-dimethoxyphenyl)-3-oxobutyramide] color pigment Yellow 83 and Yellow 108,300 were purchased from Heubach Color Pvt. Ltd. (Gujarat, India). Pigment Red 300,203 was purchased from Heubach Color Pvt. Ltd. (India). Titanium dioxide R-2196 was purchased from Wuxi Haopu Titanium Industry Co., Ltd. (Gansu, China). Carbon was purchased from Jung-Woo Co., Ltd. (Seoul, Republic of Korea). DISPERBYK-111 was used as a dispersant and was purchased from BYK-Chemie GmbH Co., Ltd. (Wesel, Germany).

### 2.2. Methods

Ten experimental composite resins were prepared as per the compositions listed in Table 1. All the resins were mixed with 4% TPO as a photo-initiator at 405 nm.

#### 2.2.1. Measurement of the Viscosity

The viscosities of the resins were determined using a viscometer (DV2TLV, Rheometer, Brookfield) at a constant volume of 100 mL. The temperature of the specimens was maintained in the range of 22–25 °C. The readings were recorded at 10 rpm for 2 min, and the result was recorded in centipoise. Each specimen was measured five times, and the average value was calculated. Considering the data for the ten resins, viscosities of less than 2000 cp were chosen. The specimens were printed using a custom UV–SLA printer equipped with a projector (3D printer A+ type, Shindoh, Republic of Korea).

#### 2.2.2. Measurement of the Degree of Conversion (DC)

The DC of the resins was determined using Fourier transform infrared (FT-IR) spectroscopy (Bruker Optik Gm bH (Vertex-70V/Hyperion 3000)) in the total reduced reflectance mode. Specimens with standard dimensions (diameter = 15 mm; thickness = 1 mm; *n* = 5 per resin) were prepared from each resin. The specimens were analyzed in the range of 3500–500 cm^−1^ at a speed of 4 cm^−1^ using an FT-IR spectrometer in the attenuated total reflection (ATR) mode. A total of 32 scans were performed [33,34,35].

#### 2.2.3. Measurement of the Flexural Strength

The flexural strength was determined using a 3-point bending test (*n* = 20) following the ISO 4049:2000 standard. A 3D printer was used to print rectangular specimens (1 cm × 8 cm × 0.5 cm). The printed specimens were placed in isopropyl alcohol (IPA) and washed for 10 min. Subsequently, they were cured for 5 min using a UV machine. The flexural strength test was performed using a universal testing machine (UTM, Shimadzu corporation (AGS-X)). The specimens were immersed in a 37 °C water bath for 24 h, and their thicknesses and widths were measured. The specimens were measured using the 3-point jig of the universal testing machine (measurement conditions: speed of 1000 mm/min and a load of 10 kN). The measurements were performed three times under the same conditions, and the average value was calculated.

#### 2.2.4. Measurement of the Hardness

The hardness of the monomer resins with different ratios was used to confirm their surface strength. The specimens were manufactured using polymer resin according to the specifications (2 × 2 × 2 cm; *n* = 3) of a 3D printer. The specimens manufactured using the 3D printer were washed and soaked in IPA to remove the unreacted monomers. After removing the remaining unreacted monomer from the surface, the specimens were cured using a UV machine for approximately 5 min to polymerize the unreacted monomer and subsequently aged in an oven at 90 °C for approximately 10 h. The hardness was measured using a Durometer Digital Shore A system (Zwick Roell (Zwcik 3130, ZwickRoell GmbH&Co. KG, Ulm, Germany)). The measurement was conducted for 0 to 99 s with a 10 N force using a truncated cone opening angle of 35°. The specimens were tested three times under the same conditions, and the average value was obtained.

#### 2.2.5. Measurement of the Compression Strength

The compression specimens were manufactured using polymer resin according to the specifications (column diameter d = 15 ± 0.5 mm and length L = 40 ± 1 mm) of a 3D printer. The specimens manufactured using the 3D printer were washed and soaked in IPA to remove the unreacted monomers. After removing the remaining unreacted monomer from the surface, the specimens were cured using a UV machine for approximately 5 min to polymerize the unreacted monomer. The compression was measured using a Zwick/Roell (Z150) tester (ZwickRoell GmbH & Co. KG, Ulm, Germany). The measurements followed the test method specified in the ISO 604 standard. The test parameters were preloading = 100 N, test load = 10 mm/min, and maximum deformation = 15%. The tests were focused on the comparison and analysis of the compression strength and compression modulus.

#### 2.2.6. Measurement of the Tensile Strength

The tensile strength specimens were manufactured using according to the ISO 527-2 specifications (full length L_3_ = 150 mm, parallel length L_2_ = 60.0 ± 5 mm, gauge length L_1_ = 108 ± 1.6 mm, thickness h = 4.0 ± 2 mm) of a 3D printer. The specimens manufactured using the 3D printer were post-processed in the same way as the compressed method. The tensile strength was measured using a Zwick/Roell (Z150) tester (ZwickRoell GmbH&Co. KG, Ulm, Germany). The test parameters were test load = 1 mm/min and maximum deformation = 1%/min. The tests were focused on the analysis and comparison of the tensile strength and Young’s modulus.

#### 2.2.7. Measurement of the Absorption and Solubility

A computer-aided design (CAD) program was used to create a circular film with a thickness of 1.5 mm and a diameter of 10 mm. To remove the unreacted resin, all the specimens were exposed to the same amount of light and immersed in IPA for 5 min. The specimens were cured for 5 min under a UV machine before drying in the oven at 25 °C for 24 h. The beaker containing the specimens and water was placed in the oven at 25 °C, and the specimens were maintained under this condition for 7 days before conducting weight measurements. Measurements of the water absorption and solubility were performed according to the ISO 4049 standard.

## 3. Results and Discussion

Photocurable 3D printer resins can be created using Bis-EMA and TMPTMA. The printed products exhibit excellent mechanical properties, resolution, and stability. Herein, an SLA-3D printer was employed. SLA technology is the oldest 3D printing technology in use. The SLA method was chosen because it is faster and more precise than the field-flow fractionation (FFF) method. The SLA-3D printer prints structures while creating layers. Therefore, if the viscosity of the resin is extremely high or low, the resolution of the structure will be reduced [2].

In 3D printer composite resins, Bis-GMA is the most commonly used methacrylate material. However, its applications are limited by its high viscosity. Therefore, Bis-EMA, which has a structure and properties similar to those of Bis-GMA, was chosen for this study. Bis-EMA consists of a rigid bisphenol A core and flexible ethoxylated side chains. TMPTMA is a flexible, low-viscosity, diluent monomer used to achieve a high DC and the homogeneous dispersion of monomers. Therefore, to control the viscosity of the resin, TMPTMA was added at a ratio of 10% of the total weight. However, the excess amount of TMPTMA can deteriorate the physical properties and viscosity of the resin [36].

When the viscosity of the resin mixture exceeded 2000 cp in the preliminary test, missing layers appeared in the resin output. To supplement the resin, the viscosity of the resin was limited to less than 2000 cp. To initiate the photopolymerization reaction, a photo-initiator with an absorption peak in the range of 350–420 nm must be added to the colorless resin. Camphorquinone (CQ) is the most commonly used photo-initiator. However, in 3D printing, the polymerization efficiency and the use of yellow-colored initiators may have limitations. Furthermore, in the case of CQ, the visible light absorption peak range was 380–520 nm, and the polymerization efficiency was low. However, TPO has an absorption peak range of 380–425 nm and a high polymerization efficiency. When exposed to UV light, the color of the initiator changes from yellow to transparent [37].

### 3.1. Viscosity of the Resin

The polymerization rate of a resin is correlated with its initial viscosity, because it affects the mobility of monomers and free radicals. We performed a viscosity analysis of the resins. The viscosity of the resins showed different trends depending on the contents of the urethane oligomer and diluent monomer (Table 1). For the synthesizing resins B100 to O100, the weight ratio of Bis-EMA was decreased from 100 to 0%, while the weight ratio of the oligomers was increased from 0% to 100%. As shown in Figure 2, as the weight ratio of Bis-EMA decreased and the weight ratio of the oligomers increased, the viscosity increased from 659 to 5980. Comparing the viscosity values of O100 in resin B100, a difference of approximately 90% was observed. A resin with a high viscosity, such as O100, is not suitable for 3D printing because it reduces the performance of the resin. As a co-monomer, TMPTMA, with a low viscosity, was added to obtain a viscosity suitable for a 3D printing resin. Because of its low viscosity, TMPTMA is used as a diluent. The viscosity of the resin decreased by approximately 30–50% depending on the TMPTMA content. With the addition of TMPTMA, the viscosity of the resin with 10% oligomer decreased by only 10%. The viscosity of the resin with 30% or more oligomer decreased by approximately half depending on the additional amount of TMPTMA. When 10% of the diluent monomer, TMPTMA, was added to the blended resin, its viscosity decreased. Herein, seven resins with viscosities of less than 2000 cp were chosen from a total of ten resins (Table 2).

### 3.2. Dependance of DC on the Resin Content

Bis-EMA, TMPTMA, and urethane oligomers are di-functional monomers. In urethane oligomers, hydrogen bonds formed by urethane proton donor groups can enhance the mechanical properties of the resin. Owing to their lower flexibility, urethane oligomers exhibit a lower degree of conversion (DC) and higher conformational homogeneity than Bis-EMA. Therefore, unreacted monomers exist in photopolymerized resins with a low DC. The DC range of the 3D-printed cubes is 30–50%. Unreacted monomers may exist in the cubic specimen. When the unreacted monomer enters the body, it causes cell death. This phenomenon necessitates the cytotoxicity testing of the prepared materials. Unreacted monomers should be removed post-curing. Therefore, the remaining resin on the surface was removed, and post-polymerization was performed using a UV machine. The remaining uncured monomer was sufficiently polymerized [14].

Strong hydrogen bonds within the resin lead to an increase in the DC. The DC of the resin is also determined by the Bis-EMA content and the mechanical properties of the composite resin [35,37]. In addition, composite resins with a lower viscosity and higher flexibility of the monomer molecules exhibit a higher DC owing to the free radical mobility of the monomer and the growing polymer chain. Therefore, the absence of –OH groups in Bis-EMA increased the flexibility and DC value of the resin. Increasing the DC value has the effect of discharging the remaining photo-initiator in the structure. As the majority of the monomers in the organic phase of the 3D printer resin were dimethacrylate, an absorption band of 1635–1640 cm^−1^ (corresponding to the C=C of the methacrylate group) was used to quantify the unreacted methacrylate groups (Figure 3). The peak intensity was compared to that of an internal standard that did not participate in the polymerization reaction, namely, the C–C aromatic absorption band at 1608–1610 cm^−1^. The absorbance ratios of C=C (aliphatic) and C–C (aromatic) before and after curing were calculated using the baseline peak in the range of 1571.7–1660.41 cm^−1^ to determine the unreacted methacrylate group (Appendix A). Fourier transform infrared spectroscopy (FT-IR) was used to monitor the UV curing of the resin during the SLA process. In addition, the following equations were used to calculate the ratio of the monomer to the polymer double-bond content or the DC of residual bonds in the resin [35]:

DC = (1 − [A-cured/A-uncured]) × 10
(1)

where A-cured represents the proportion of aliphatic (1639 cm^−1^) to aromatic (1608 cm^−1^) C=C peak area of the cured resin. A-uncured represents the equivalence ratio of the resin before polymerization [34].

The DC degree of a cube printed using the 3D printer is shown in Figure 4. B2T1O7 exhibited the highest DC value (54.36%) among the seven selected resins.

### 3.3. Dependance of Mechanical Properties on Resin Content

An increase in the concentration of oligomers in the resin affects the flexural strength. The addition of urethane oligomers to the resin increases the intermolecular interactions and hydrogen bonding, which can affect the flexural strength and modulus of the crosslinked methacrylate [38]. Our analysis shows that the flexural strength ranged from 40 to 60 MPa (Table 3). B2T7O1 exhibited the highest flexural strength, while B9O1 presented the lowest value. The resins containing 10% TMPTMA demonstrated higher flexural strengths than those containing high amounts of the oligomer (Figure 4). The resin printed via 3D printing exhibited a flexural strength of 60–80 MPa, which was comparable to those of the PMMA and acrylic-based resins [39]. In other words, the as-prepared resin is suitable for medical and dental applications (temporary restorations, denture bases, etc.).

The 3D-printed cube with B9O1 exhibited the highest Vickers hardness of 57.8 ± 1.2 MV, and the lowest surface hardness of 41.6 ± 0.6 MV was observed in B2T1O7 (Figure 5). The surface hardness of the resin containing a greater oligomer content was lower than that of the resins containing 10% TMPTMA.

The values of the compression and compression modulus were the highest for the B100 specimen at 28.2 MPa and 733.2 MPa and the lowest for the B2T1O7 specimen at 15.6 MPa and 246.7 MPa. The values of the compression and compression modulus showed similar results (Figure 6).

The tensile strength of the B8T1O1 specimen was the highest at 14.5 MPa, and the B2T1O7 specimen was the lowest at 3.3 MPa. The Young’s modulus was the highest at 203.2 MPa in the B6T1O3 specimen(Figure 7). The measured values of the B6T1O3 and B8T1O1 specimens showed similar values, with a difference of about 8.1 MPa. Both specimens had low oligomer contents and contain TMPTMA. The B2T1O7 specimen showed the lowest value at 48.9 MPa. In the case of the tensile strength, it was found that the values changed according to the Bis-EMA and TMPTMA contained in the specimen.

The studied mechanical properties tended to vary depending on the DC value and content ratio of the resin. The 3D-printed cube with B9O1 exhibited the highest Vickers hardness of 57.8 ± 1.2 MV, and the lowest surface hardness of 41.6 ± 0.6 MV was observed in B2T1O7. Additionally, B2T1O7 showed low values in its compressive strength, compressive modulus, tensile strength, and Young’s modulus. B2T1O7 exhibited the highest DC and flexural strengths but the lowest surface hardness, compression, and tensile strength. These findings indicate the presence of TMPTMA and lower amounts of Bis-EMA. Because the urethane oligomer has a flexible ester bond in the resin, it can significantly reduce the hardness compared to the rigid benzoic ring of Bis-EMA [33,37].

### 3.4. Water Stability of the Resins

The hydrophobicity of the monomers is one of the most important factors for predicting the water adsorption of the resin. In addition, the susceptibility to water absorption and solubility of the resin is related to its DC value. The –OH groups in the resin structure can form hydrogen bonds with water molecules and, thus, the proportion of each monomer in the resin determines the water absorbance capacity. The Bis-EMA structure lacks –OH groups, and resins containing a high proportion of this monomer exhibited a lower water absorption. The highest water absorbance value was observed for B2T1O7, with similar values observed for the other selected formulations (Table 4). Structures printed using a 3D printer should not corrode when exposed to air or water at 24 °C room temperature. However, the structures with a low stability are easily damaged by the external environment. The accumulated damage eventually corrodes or deforms the structure. A low stability indicates that the structures are affected by the light emitted by the 3D printer. A circular specimen was printed to test the stability of the synthesized resin. The degree of damage caused by air and water exposure was measured using the water solubility and absorption properties of the specimen [14]. All the specimens showed considerable water solubility and absorption (Figure 8). The water solubility (W_sol_) and absorption (W_abs_) were calculated using the following equations:
W_sol_ = (M_a_ − M_c_)/V
(2)


W_abs_ = (M_b_ − M_c_)/V
(3)

where M_a_ is the mass of the specimen after drying to a constant weight prior to water immersion; M_b_ is the mass of the specimen after immersion in water for a specific time; and M_c_ is the mass of the reconstructed specimen after drying until a constant weight is achieved. V is the volume (mm^3^) of the specimen. The sensitivity of the resin to water was analyzed using two methods. The results showed that the water absorption and solubility were affected by both the monomer content and the DC value of the resin.

The resins containing –OH groups exhibited higher water absorption values. The absence of a hydroxyl group in the ethoxylated monomer (Bis-EMA) results in reduced water absorption [40,41,42]. Therefore, the B100 resin showed a low absorption value. As a result, water molecules can cleave intermolecular forces and diffuse within the polymer chain to leach unreacted monomers. Solubility and adsorption lead to the hygroscopic expansion, plasticization, and hydrolysis of the resin, which weaken its mechanical properties over time. Therefore, understanding the behavior of the resin in an aqueous environment is important for evaluating its properties [43,44].

### 3.5. Changes in the Physical Properties Based on Additives

The addition of a pigment to a transparent resin alters the light absorption rate of the 3D printer. Light absorption and transmittance vary depending on the transparency of the resin. When the light absorption rate changes, the amount of light irradiated by the 3D printer on the resin changes, influencing the structure’s resolution. Therefore, it is necessary to check the resolution of the resin based on the additive content. Herein, three patterns considering the column height, circle size, and line were designed using a CAD program to test the resolution of the synthesized resin. Furthermore, the resin showing the optimal performance, B6T1O3, was chosen as the base resin. The base resin was thoroughly mixed with white (titanium dioxide) and black (carbon black) pigments before being filtered through filter paper. As shown in Figure 9a, the column pattern was a series of vertical bars that increase in height from 5 to 50 mm in 5 mm increments. The circle diameter was designed to range from 0.6 to 6 mm (Figure 9b). As shown in Figure 9e,f, it was increased in steps of 0.1 mm from 0.6 to 1 mm and then by 1 mm from 1 to 6 mm, respectively. The line (Figure 9c) was designed to range from 0.2 to 5 mm. The line increased in steps of 0.1 mm from 0.2 to 1 mm and then by 1 mm from 1 to 5 mm. All the patterns were observed by comparing the resolution by adding a pigment to the B6T1O3 resin. As shown in Figure 9, B6T1O3 shows a good resolution as a 3D printer resin, with very small printing patterns of a size of 0.2 mm. Additionally, detailed measurement information for the output resolution is summarized in Table 5.

Depending on the application of the 3D printer resin, various additives are used. Inorganic and organic pigments are used as pigment additives. Organic pigments develop color faster than inorganic pigments. However, with prolonged use, the performance stability of the resin deteriorates. Furthermore, organic pigments are typically fluorescent. However, inorganic pigments are commonly used as additives for 3D printer resins. Inorganic pigments offer more color variation than organic pigments. Therefore, resins of various colors can be manufactured with a high stability. However, when selecting a pigment additive, the pigment formulation and particle size must be considered. Even when the same amount of pigment is added, various colors are obtained depending on the particle size. That is, the pigment dispersion rate varies according to the particle size. If the pigment is not sufficiently dispersed in the resin, the pigment particles may reflect or absorb the UV radiation emitted by the 3D printer. SLA-style 3D printers print structures one layer at a time. Undispersed particle fragments prevent the structure’s accumulation. This can result in cracks or a lack of hardening of the structure.

The effects of the additive content and ratio on the resin resolution were examined by preparing five different pigmented resins (Table 6). The Eiffel Tower, with its many thin lines and complexities, was selected as the model structure and printed using all five resins (Figure 10). All the resin structures demonstrated a good resolution, without cracks. The transparency of the resin varied depending on the pigment content, and the microstructure of the Eiffel Tower was more clearly visible when it was printed using a less transparent and more saturated resin.

Resin has a wide range of applications in 3D printing and must be optimized for each condition and environment. Fields such as crafts, where detailed structures are necessary, require a high resolution and saturation. In medical applications, the printing conditions depend on whether the printed structures are designed to be placed inside or outside the human body. Toxicity and biocompatibility testing are also critical when using 3D-printed structures in medical applications. Industrial applications require excellent mechanical properties to withstand pressure and impacts [32,45].

It is necessary to test the mechanical properties of the resin after the addition of pigments. Furthermore, the resolution and stability of a pigmented resin may vary over time due to precipitation. Therefore, the precipitation of the resin and the manufacturing period must be considered when performing stability and resolution tests. Toxicity and biocompatibility tests should be conducted with and without pigment addition and by varying the monomer resin ratio [26,28,46].

## 4. Conclusions

In this study, the influences of the individual components on the properties of the 3D printer resin were examined. The mechanical and physical properties of the Bis-EMA, TMPTMA, and urethane oligomers were analyzed, and it was found that the physical and mechanical properties of the resin were affected by the monomer content ratio. The three test patterns showed a good printer output resolution. Using the five pigmented resins, the Eiffel Tower pattern was successfully printed without cracks. This demonstrated the performance and potential of acrylate-based mixed resins for use in the rapidly growing 3D printing market. Research on material selection and mixing should be further expanded so as to keep pace with the evolving technology. The characteristics of the materials used in 3D printing can be improved through better control of the material properties. Before the commercialization of the as-developed composite resins can proceed, additional studies are required, including further examination of the resin and pigment variables, biocompatibility tests, and stability tests. 

## Figures and Tables

**Figure 1 polymers-14-05198-f001:**
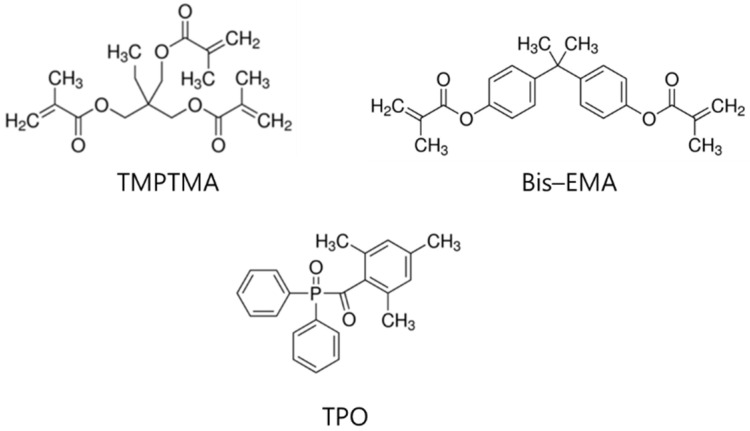
Chemical structure of the monomers (Bis-EMA and TMPTMA) and initiator (TPO).

**Figure 2 polymers-14-05198-f002:**
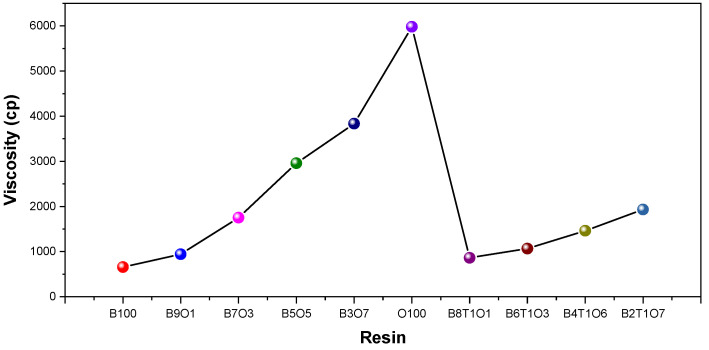
Viscosity of various 3D printing resins.

**Figure 3 polymers-14-05198-f003:**
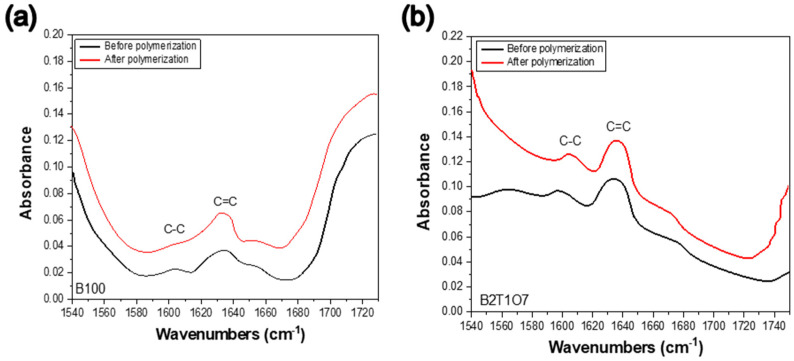
FT-IR spectra before and after light polymerization showing typical C−C and C=C peaks by. (**a**) B100, (**b**) B2T1O7.

**Figure 4 polymers-14-05198-f004:**
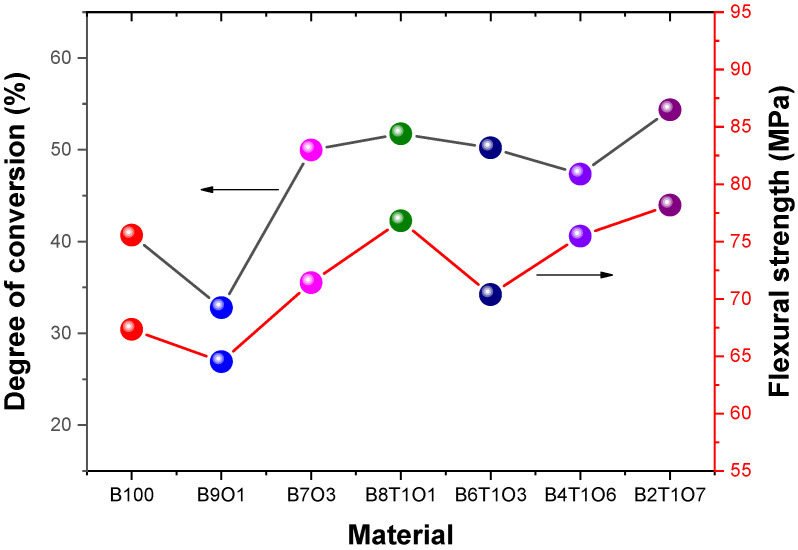
Degree of conversion and flexural strength of the 3D-printed specimens.

**Figure 5 polymers-14-05198-f005:**
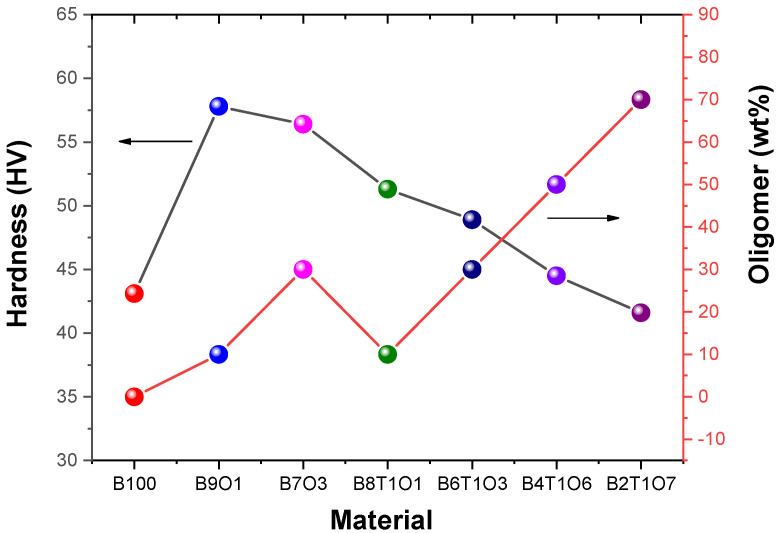
Surface hardness of the 3D-printed specimens.

**Figure 6 polymers-14-05198-f006:**
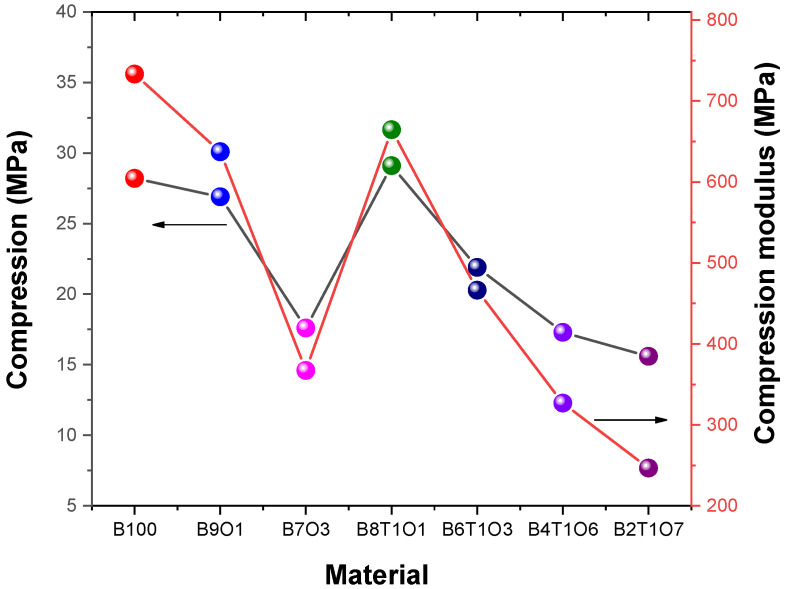
Compression and compression modulus of the 3D-printed specimens.

**Figure 7 polymers-14-05198-f007:**
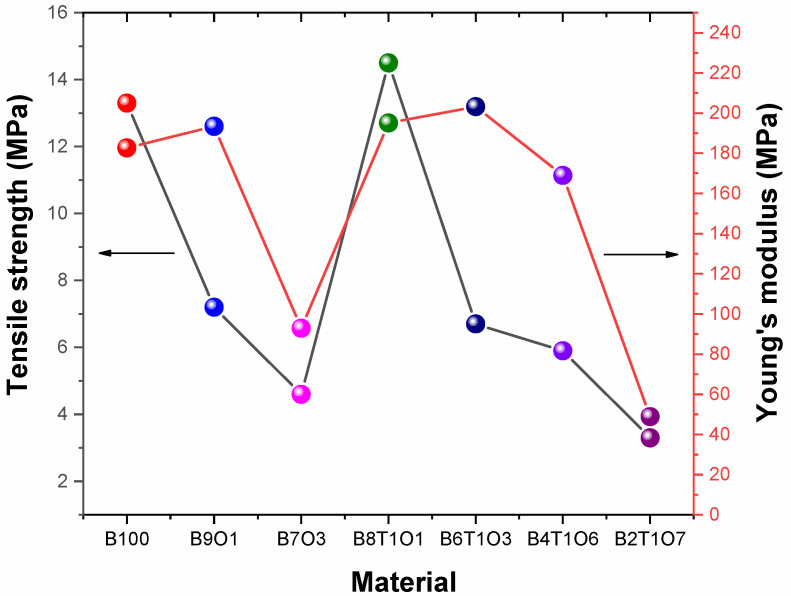
Tensile strength and Young’s modulus of the 3D-printed specimens.

**Figure 8 polymers-14-05198-f008:**
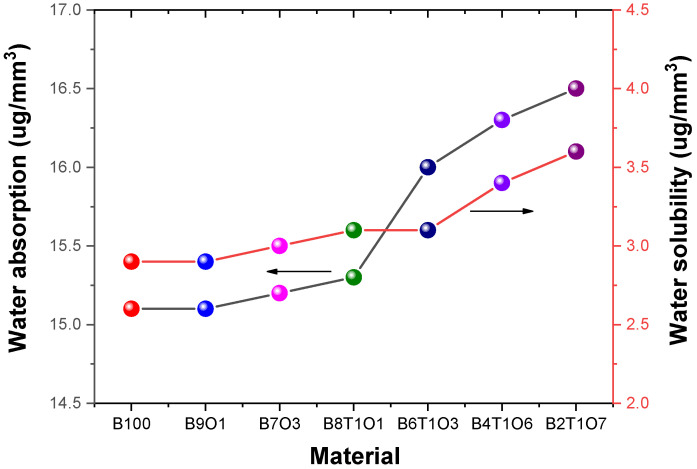
Water absorption and water solubility of the 3D-printed specimens.

**Figure 9 polymers-14-05198-f009:**
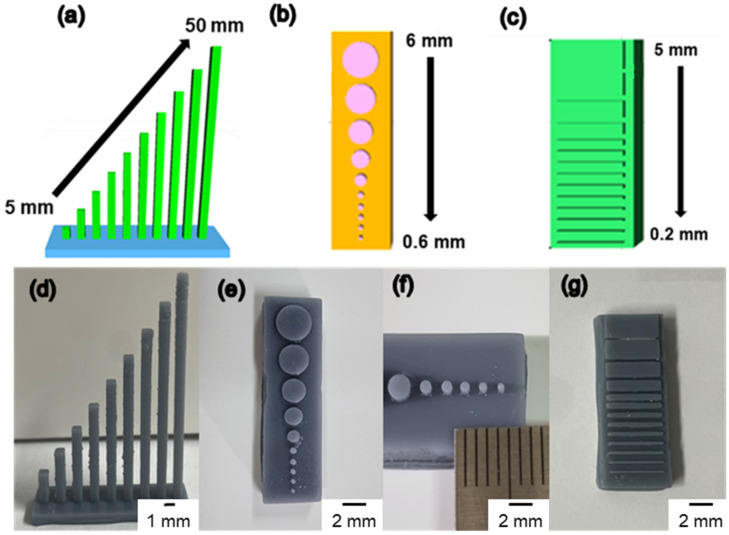
Comparison of resin resolution based on CAD photograph. (**a**) High-aspect-ratio column pattern. (**b**) Circle pattern. (**c**) Line pattern. Three structure patterns were printed with a 3D printer using a gray color resin. (**d**) Column structure (aspect). (**e**) Circle structure (front). (**f**) Magnified image with a scale range of 0.6–1.0 mm. (**g**) Line structure (front).

**Figure 10 polymers-14-05198-f010:**
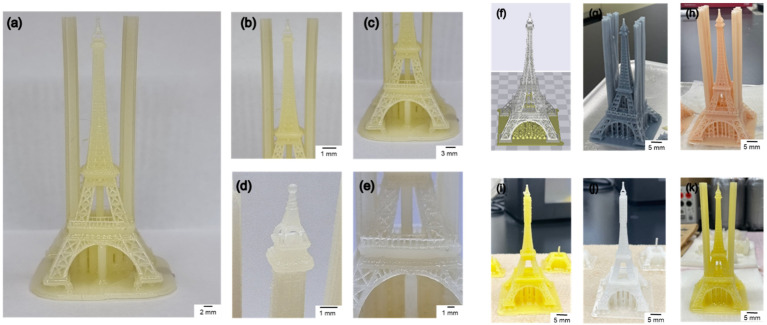
Eiffel Tower model printed by a 3D printer with different resin colors. (**a**) Eiffel Tower printed using yellow color resin; (**b**) Eiffel Tower top part; (**c**) Eiffel Tower bottom part; (**d**) Eiffel Tower top vertex; (**e**) Eiffel Tower middle part; (**f**) Eiffel Tower 3D CAD model; Eiffel Tower model printed using (**g**) gray color resin, (**h**) pink color resin, (**i**) fluorescent resin, (**j**) white color resin, and (**k**) yellow color resin.

**Table 1 polymers-14-05198-t001:** Urethane-acrylate resin material contents used herein.

Name	Composition (wt%)			Viscosity (cp)
	Bis-EMA	TMPTMA	Oligomer	
B100	100	0	0	659
B9O1	90	0	10	942
B7O3	70	0	30	1752
B5O5	50	0	50	2958
B3O7	30	0	70	3834
O100	0	0	100	5980
B8T1O1	80	10	10	864
B6T1O3	60	10	30	1068
B4T1O5	40	10	50	1464
B2T1O7	20	10	70	1932

**Table 2 polymers-14-05198-t002:** Composition and degree of conversion of urethane acrylate resin materials.

Name	Composition (wt%)			Viscosity (cp)	Degree of Conversion (DC, %)
	Bis-EMA	TMPTMA	Oligomer		
B100	100	0	0	659	40.68
B9O1	90	0	10	942	32.79
B7O3	70	0	30	1752	49.97
B8T1O1	80	10	10	864	51.74
B6T1O3	60	10	30	1068	56.21
B4T1O5	40	10	50	1454	47.34
B2T1O7	20	10	70	1932	54.36

**Table 3 polymers-14-05198-t003:** Mechanical properties of urethane acrylate resin materials.

Name	Flexural Strength (MPa)	Hardness (HV)	Compression (MPa)	Compression Modulus (MPa)	Tensile Strength (MPa)	Young’s Modulus (MPa)
B100	67.34	43.1	28.2	733.2	13.3	182.7
B9O1	64.52	57.8	26.9	637.2	7.2	193.4
B7O3	71.41	56.4	17.6	367.2	4.6	92.9
B8T1O1	76.80	51.3	29.1	664.4	14.5	195.1
B6T1O3	70.37	48.9	21.9	466.3	6.7	203.2
B4T1O5	75.47	44.5	17.3	327.0	5.9	169.0
B2T1O7	78.17	41.6	15.6	246.7	3.3	48.9

**Table 4 polymers-14-05198-t004:** Water absorption and solubility of the urethane acrylate resin materials.

Name	Thickness (mm)	Water Absorption (µg/mm^3^)	Water Solubility (µg/mm^3^)
B100	1.52	15.1	2.9
B9O1	1.51	15.1	2.9
B7O3	1.51	15.2	3.0
B8T1O1	1.51	15.3	3.1
B6T1O3	1.52	16.0	3.1
B4T1O5	1.51	16.3	3.4
B2T1O7	1.53	16.5	3.6

**Table 5 polymers-14-05198-t005:** Summary of the resin resolution, calculated by CAD photograph (a) Column, (b) Circle, and (c) Line structure.

Model Type	Column (mm)	Circle (mm)	Line (mm)
CAD model	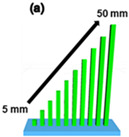	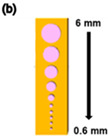	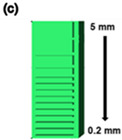
Model size			0.2
	5	0.6	0.3
	10	0.7	0.4
	15	0.8	0.5
	20	0.9	0.6
	25	10.0	0.7
	30	20.0	0.8
	35	30.0	0.9
	40	40.0	10.0
	45	50.0	20.0
	50	60.0	30.0
			40.0
			50.0
Size of printed structure			0.21
	4.98	0.61	0.30
	10.01	0.70	0.40
	14.99	0.80	0.51
	20.00	0.89	0.60
	25.01	10.00	0.70
	29.99	20.01	0.80
	35.00	30.00	0.91
	40.01	40.01	10.00
	45.01	50.00	20.01
	50. 00	60.01	29.99
			40.02
			50.00

**Table 6 polymers-14-05198-t006:** Pigment ratio of urethane acrylate resin.

Resin (Color) *	Base Polymer B6T1O3 (g)	Pigment (g)
G (Grey)	300	0.283
H (Pink)	300	0.294
I (Fluorescent)	300	0.288
J (White)	300	0.280
K (Yellow)	300	0.281

* The grey region contained titanium dioxide (0.28 g) and carbon black (0.003 g). The pink resin contained titanium dioxide (0.28 g) and red 300,203 (0.014); the fluorescent resin contained titanium dioxide (0.28 g) and yellow 108,300 (0.008 g); and the white resin contained only titanium dioxide (0.28 g). Furthermore, the yellow resin contained titanium dioxide (0.28 g) and yellow 83 (0.008 g).

## Data Availability

Not applicable.

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
