# Peer review of "Development of Bisphenol-A-Glycidyl-Methacrylate- and Trimethylolpropane-Triacrylate-Based Stereolithography 3D Printing Materials"

_polymers, 2022, doi:10.3390/polym14235198_

Round 1

Author Response

Please find our response in the attachment

Reviewer 2 Report

The paper : Development of bisphenol-A glycidyl methacrylate and trimethylolpropane triacrylate based stereolithography 3D printing materials presents some interesting results in a requested field of 3D printing. The paper is well written with many experiments and results. The references section is well prepared. 

Use a dot after: Many parameters, such as the mechanical properties (flexural strength, hardness, etc.) and physical properties (viscosity, color, etc.), must be considered in the selection and formulation of 3D printing resin materials and additives [28-30] 

why Eiffel Tower and not a stent ? 

l237: in figure Degree of converion not conversin 

L271: eq. (1) 

line 299: Among - among 

in figure 6 mention the scale , on the picture of the ruler  and a scale on d), e) and g) 

line 333: number for equations 

re-structure 4.Conclusions section 

what is this section: 5. Patents ? 

Author Response

(The authors gave the same response as above.)

Reviewer 3 Report

The authors investigated a novel composite resin containing commonly used photocurable polymers in different aspects, including viscosity, flexural strength, hardness, water absorption, solubility, and printing fidelity. The manuscript was well-designed, structured, and prepared. The viscosity, water solubility, precipitation, hardness, and flexural strength are of researchers' interest in the photocurable resin field. The final prints showed good print fidelity and aligned with the initial hypothesis.

However, some issues could be addressed by the author as listed below:

· The impact of your work remained unclear. The authors claimed that the resins possess good physical and mechanical properties and printing resolution. However, the applications of the composite resin were not described in detail. Line 101-103 briefly summarizes its future applications in fields that required good mechanical strength and etc. I would suggest the authors dig out in what specific areas the resins you developed may play a vital role, and provide details on how the applications align with your resin properties.

· The focus of material characterization was on mechanical strength, however, some fundamental experiments should be added such as compression and universal tensile tests. These tests would give essential properties of your resins, i.e. compressive modulus and young's modulus. SLA printing could easily make standard specimens for compression and tensile tests, these two tests should be added.

· The authors mentioned the resin could potentially be used in dental and medical printing. In that case, basic toxicity (MTT assay) and biocompatibility tests (Calcein AM/PI cells staining) should be granted. However, it is understandable that the authors may prefer to focus on mechanical characterization. I would suggest the authors look for some references that can strongly support your resin's applications in medical scenarios.

· The authors should also describe how the printing process was, especially in the final product printing, no details were given.

· In Figure 2, the viscosity of different resins was given. However, the way the author presents may confuse the readers. The group names could be given in legends rather than independent data points. The way the curve was sketched suggested an increase in viscosity by adding oligomer, I would suggest using either Bis-EMA or Oligmoer concentration as the x-axis and using a clear legend to distinguish the groups. 

· Line 329-330, only if replicate experiments were done and data were given that one can claim there is no statistically significant difference. I believe there is no evidence showing replicate experiment data.

· Line 345-346, the authors need further references to support this argument.

· Figure 6 (d,e,f,g) needed scale bars instead of a picture of a ruler without scales. The printing resolution should be quantitatively represented, for example, the authors could summarize the resin resolution in table format and calculate printing errors from the comparison of prints & models.

· Line 405-409, reference needed.

· Line 411-417, the writing of this paragraph was convoluted and misleading as therefore/furthermore was used twice. Also, the tone of the paragraph implied criticism of your own work, which I believe you may want to switch your tone to a future perspective of upcoming work you plan to do so.

· Line 449, a typo?

Author Response

(The authors gave the same response as above.)

Round 2

Reviewer 1 Report

The authors have put their best efforts into improving the quality of the manuscript. All the comments are well addressed. The revised version of the manuscript can be accepted for publication.

Reviewer 2 Report

The article can be published in the current state , the authors had perform all the minor corrections requested.